# HDAC6 Facilitates PRV and VSV Infection by Inhibiting Type I Interferon Production

**DOI:** 10.3390/v17010090

**Published:** 2025-01-13

**Authors:** Hu Zheng, Xiaohui Yang, Haiwen Zhong, Changxu Song, Zhenfang Wu, Huaqiang Yang

**Affiliations:** 1State Key Laboratory of Swine and Poultry Breeding Industry, National Engineering Research Center for Breeding Swine Industry, College of Animal Science, South China Agricultural University, Guangzhou 510642, China; zhenghu@stu.scau.edu.cn (H.Z.); yangxiaohui@stu.scau.edu.cn (X.Y.); hw180214@stu.scau.edu.cn (H.Z.); cxsong@scau.edu.cn (C.S.); 2Yunfu Branch Center of Guangdong Laboratory of Lingnan Modern Agricultural Science and Technology, Yunfu 527400, China

**Keywords:** HDAC6, PRV, VSV, Type I IFN, antiviral immunity, host-virus interaction

## Abstract

HDAC6 modulates viral infection through diverse mechanisms. Here, we investigated the role of HDAC6 in influencing viral infection in pig cells with the aim of exploiting the potential antiviral gene targets in pigs. Using gene knockout and overexpression strategies, we found that HDAC6 knockout greatly reduced PRV and VSV infectivity, whereas HDAC6 overexpression increased their infectivity in PK15 cells. Mechanistic studies identified HDAC6 as a DNA damage inhibitor in PK15 cells. HDAC6 overexpression attenuated DNA damage levels, which can further reduce type I IFN production to promote viral infection. Conversely, HDAC6 deficiency can limit viral infection by increasing DNA damage-mediated type I IFN production. This work demonstrates that HDAC6 affects the infection process of multiple viruses by modulating type I IFN production, highlighting a regulatory role of HDAC6 linking host immune response and viral infection levels in pig cells.

## 1. Introduction

Histone deacetylase 6 (HDAC6) is a unique HDAC family member among the 18 mammalian HDACs. HDAC6 is a class II deacetylase that is predominately located in the cytoplasm and acts on non-histone protein substrates [1,2,3]. HDAC6 has the N-terminal nuclear export signal (NES) and nuclear localization signal (NLS). HDAC6 is mainly present in the cytosol as the NES has a stronger signal than NLS [4]. It has two adjacent catalytic domains (CD), CD1 and CD2, which show deacetylase activity and ubiquitin E3 ligase activity [5,6]. The C terminal of HDAC6 contains a ubiquitin-binding domain BUZ mediating ubiquitin recruitment to stimulate aggresome formation [7,8]. The structural feature highlights that HDAC6 acts on a wide range of targets and regulates multiple biological processes through enzymatic and non-enzymatic functions.

Effects of acetylation and deacetylation on cellular and viral proteins influence the viral infection process in host cells. HDAC6 is an acetylation eraser that plays a key role in viral infection and innate immunity [3,4]. HDAC6 has been reported to either promote or inhibit the infection process of various viruses by affecting viral life cycle stages and host antiviral response through deacetylation of diverse protein substrates and prevention of protein aggregation [9,10]. For example, HDAC6 has been reported to affect influenza A virus (IAV) infection via diversified mechanisms. HDAC6 deacetylates cytoskeleton microtubules to prevent the trafficking of viral components toward the plasma membrane of the host cell [11]. HDAC6 can also deacetylate polymerase acidic protein, which reduces IAV RNA polymerase activity to exhibit anti-IAV activity [12]. On the contrary, IAV takes advantage of the host cell’s aggresome machinery by carrying the ubiquitin chain and recruiting HDAC6 to facilitate virus entry into host cells. In this respect, the ubiquitin-binding activity, but not the deacetylase activity, of HDAC6 is required for efficient viral uncoating and infection [13]. In addition to the viral life cycle, HDAC6 has been reported to regulate the host interferon (IFN) response. During infection with Sendai virus, HDAC6 deacetylates β-catenin and enhances its nuclear translocation and promoter binding, acting as a co-activator for IRF3-mediated transcription of type I IFN [14]. Choi et al. report that HDAC6 activates RIG-I-mediated viral RNA sensing and type I IFN induction through deacetylation of RIG-I [15]. Together, these works demonstrate that HDAC6 confers its antiviral immunity by upregulating the host type I IFN response. Interestingly, Wang et al. report that HDAC6 deletion enhances poly(I:C)-induced INF-β expression in macrophages, suggesting an inhibitory role of HDAC6 in antiviral innate immune responses [16]. HDAC6 plays a dual role in viral infection and pathogenesis. HDAC6 can either help host cells defend against the virus or be hijacked by the virus to aid its infection.

The complex effect of HDAC6 highlights its diverse functions in regulating viral infection under different circumstances. To investigate the species-specific HDAC6 effect on viral infection, we investigated how HDAC6 affects pseudorabies virus (PRV) and vesicular stomatitis virus (VSV) infection in pig cells using HDAC6 gene knockout (KO) and overexpression methods. We also investigated the possible mechanisms underlying the modulating ability of HDAC6 on viral infection in pig cells.

## 2. Materials and Methods

### 2.1. Viruses and Cells

PRV is a virulent field strain isolated from infected pigs (GenBank accession number MT197597). VSV is an engineered virus, created by fusing protein G with EGFP in Indiana strain background. Both viruses were propagated and tittered in PK15 cells. PK15 cells were cultured in high glucose DMEM supplemented with 10% FBS and 1% Penicillin-Streptomycin (all from Gibco, Grand Island, NY, USA). During infection, PK15 cells were washed with PBS and inoculated with the virus diluted in DMEM at the specific multiplicity of infection (moi).

### 2.2. HDAC6 Gene Knockout

We chose the sequence CCGCTCCAGCCCCAATCTAG in the exon region of pig HDAC6 gnome as the guide RNA (gRNA) recognition target site. The target was synthesized as the complementary oligoes that were annealed and ligated into pX330-U6-Chimeric_BB-CBh-hSpCas9 vector (Addgene plasmid #42230) to form the gene targeting vector pX330-pHDAC6. PK15 cells were transfected with pX330-pHDAC6 using Lipofectamine 3000 reagent (Thermo Fisher Scientific, Carlsbad, CA, USA). The cells were trypsinized and seeded into 10 cm dishes at the density of 50–100 cells per dish 24 h after transfection. After culturing for approximately 2 weeks, the single cell colonies were picked up for expansion in 24-well plates. The genotypes of single cell colonies were identified by amplifying the gRNA region in the genome and sequencing to analyze the presence of base insertion/deletion (indel). The positive colonies with biallelic frameshift indels were further expanded and cytopreserved. The single cell colonies without mutation in the gRNA region in the genome were used as the wild-type non-targeting (NT) control cells.

### 2.3. Viral Quantification and Titration

For quantification of viral copies in the supernatant of cell cultures, viral nucleic acid was extracted from the supernatant using the RaPure Viral DNA/RNA Kit (Magen Biotech, Guangzhou, China). Viral DNA or RNA were quantified using Taq Pro HS Universal Probe Master Mix or HiScript II One Step qRT-PCR Probe Kit (Vazyme, Nanjing, China) in QuantStudio 5 Real-Time PCR System (Thermo Fisher Scientific, Foster City, CA, USA). The plasmids containing PRV gE and VSV L genes were used as standards to generate the standard curve to calculate the quantity of samples. The PRV quantitative PCR (qPCR) primers and probe were PRV-gE-Forward, CCCACCGCCACAAAGAACACG, PRV-gE-Reverse, GATGGGCATCGGCGACTACCTG, and PRV-gE-Probe, FAM-CGGCGCGAGCCGCCCATCGTCAC-BHQ1. The VSV RT-qPCR primers and probe were VSV-L-Forward, GATACAGTACAATTATTTTGGGAC, VSV-L-Reverse, ATGGCGTATTTGAAAGTAGAA, and VSV-L-Probe, FAM-TTGATGATGCATGATCCTGCTCTTCGT-BHQ1. Viral genome copies were expressed as Lg gE/µL and Lg L/µL for PRV and VSV, respectively, which are the Lg values of PRV gE and VSV L gene copies per µL of extracted viral nucleic acid solutions.

The viral infectivity was determined by TCID50 assay in PK15 cells. Virus was serially diluted 10-fold in DMEM, and 100 μL of each dilution was added into PK15 cells plated in 96-well plate. The cell plates were cultured for 1 week and the CPE of each well was observed and recorded. The TCID50 of virus was calculated using the Reed and Muench method.

### 2.4. Small Molecule Treatment

PK15 cells were treated with TSA or tubastatin A (MCE, Shanghai, China) for 12 h. The cells were washed with PBS and infected with PRV or VSV at moi = 0.1 or moi = 1 for 1 h. The infected cells were then washed three times with PBS and cultured in DMEM for 24 h before harvesting the cells or supernatants for analysis of cellular gene expression or viral copies, respectively.

### 2.5. Immunofluorescence

PRV-infected PK15 cells were fixed with 4% paraformaldehyde (PFA) for 20 min, permeabilized with 0.3% Triton X-100 for 10 min and blocked in 5% BSA for 1 h. To label PRV-positive cells, cells were incubated with rabbit anti-PRV polyclonal antibody (ab3534, Abcam, Cambridge, MA, USA) for 2 h at room temperature. The cells were thoroughly washed and further incubated with Alexa Fluor 488-conjugated goat anti-rabbit IgG (H+L) secondary antibody (Thermo Fisher Scientific, Carlsbad, CA, USA). DAPI was used to counterstain cell nuclei. After three washes with PBS, the positively labeled cells were observed and imaged by inverted fluorescence microscope.

### 2.6. Western Blot

Total cell protein was extracted using RIPA lysis and extraction buffer (Thermo Fisher Scientific, Rockford, IL, USA) and quantified using BCA protein assay kits (Thermo Fisher Scientific, Rockford, IL, USA). Equal amounts of protein from each sample were loaded on SDS-PAGE for protein separation and transferred to PVDF membrane (Merck Millipore, Burlington, MA, USA) for blotting. The membrane was blocked in 5% skim milk and incubated with the specific primary antibodies at 4 °C overnight. The membrane was then washed in TBST for 20 min and incubated with the HRP-conjugated secondary antibodies for 1 h at room temperature. Protein signals were visualized using an ECL detection kit (Proteintech, Wuhan, China). The protein band intensity was quantified by Image J 1.54m (NIH, Bethesda, MD, USA). The following primary antibodies were used in Western blot: HDAC6 rabbit monoclonal antibody (7558, Cell Signaling Technology, Danvers, MA, USA), H2AX rabbit monoclonal antibody (HY-P80161, MCE, Shanghai, China), γH2AX (phospho-H2AX (Ser139)) rabbit monoclonal antibody (HY-P80821, MCE, Shanghai, China), Acetyl-H2AX (Lys5) rabbit polyclonal antibody (HW093, Signalway, Greenbelt, MD, USA), and a-tubulin rabbit monoclonal antibody (PTM-5442, PTM, Hangzhou, China).

### 2.7. TUNEL Assay

Cells were fixed with 4% PFA followed by permeabilization with 0.3% Triton X-100. The fixed cells were washed twice with PBS and stained with TUNEL assay solution composed of TdT enzyme and FITC-dUTP (Beyotime, Shanghai, China) in the dark at 37 °C for 1 h. The stained cells were then thoroughly washed with PBS and observed by fluorescence microscopy.

### 2.8. AP Site Quantification

We used the DNA Damage Quantification Kit-AP Site Counting (Dojindo Laboratories, Kumamoto, Japan) to determine the abasic sites in the genomic DNA of HDAC6 KO and overexpressing PK15 cells according to the manufacturer’s instructions. Briefly, genomic DNA was labeled with Aldehyde Reactive Probe (ARP) reagent and purified by ethanol precipitation. The ARP-labeled genomic DNA and ARP-DNA standard solutions as references were allowed to bind to the 96-well plate and reacted with HRP-streptavidin solution. After thorough washing, substrate solution was added to each well for color development. The results were read on a microplate reader at OD 650 nm. The number of abasic sites in the genomic DNA was determined using the calibration curve derived from the ARP-DNA standard solutions.

### 2.9. Cell Viability Assay

Cells were seeded in 96-well plates. After attachment, the cells were treated with TSA or tubastatin A at the indicated concentrations for 12 h. The cell medium was changed with fresh medium containing 10% CCK8 reagent (Dojindo Laboratories, Kumamoto, Japan) and further cultured for 1 h. Cell viability was determined by measuring the absorbance, at 450 nm, of CCK8-stained cells using a microplate reader.

### 2.10. Statistical Analysis

All data are expressed as mean ± SD. Statistical analysis was evaluated by unpaired, two-tailed Student’s *t*-test, one-way ANOVA, or two-way ANOVA, with post hoc multiple comparison tests, as indicated in the figure legends. Results were considered significant when *p* < 0.05.

## 3. Results

### 3.1. Viral Infection Reduces HDAC6 Expression in PK15 Cells

We started by studying how HDAC6 responds to viral infection in pig cells. We used a DNA virus, PRV, and an RNA virus, VSV, separately to infect PK15 cells, a porcine kidney epithelial cell line. The infected cells were collected to determine HDAC6 mRNA expression levels in the indicated intervals. We found that both viruses caused significant reduction in HDAC6 mRNA levels in PK15 cells after infection for 12 h at moi = 1 or 18 h at moi = 0.1 (Figure 1A–D, left panels). Western blot results confirmed the reduction of HDAC6 protein upon either PRV or VSV infection (Figure 1E). It is not surprising to see reduced HDAC6 expression, as previous work has reported a broad suppression of host gene expression during PRV infection [17]. To analyze the type I IFN production, the main innate immune response defending against virus invasion, in PK15 cells, we determined the IFNβ mRNA expression in these samples and found a significantly increased IFNβ mRNA level in PK15 cells infected with PRV or VSV (Figure 1A–D, right panels).

### 3.2. HDAC6 Facilitates PRV Infection by Inhibiting Type I IFN Response in PK15 Cells

To investigate the influence of HDAC6 expression level on viral infectivity in PK15 cells, we constructed the HDAC6 KO PK15 cell lines using the CRISPR/Cas9-mediated gene targeting strategy. Through CRISPR/Cas9-mediated gene targeting and single-cell colony screen, we identified the positive cell lines with HDAC6 deficiency. The selected positive cells, KO1 and KO2, showed decreased HDAC6 expression at the mRNA level (Figure 2A) and no detectable expression at the protein level (Figure 2B). HDAC6 KO did not affect cell viability (Figure 2C). PRV infection showed mitigated cytopathic effect (CPE) in KO cells compared with wild-type control (non-targeting cells, NT). Immunofluorescence (IF) with PRV antigen-specific antibody showed decreased PRV-positive plaque numbers compared with NT (Figure 2D). We next determined the viral genome copies and viral titers in the culture supernatant and found significantly less virus in KO cells compared to NT upon infection at the moi of 0.1 and 1 (Figure 2E,F).

We explored the possible mechanism that reduces PRV infectivity in HDAC6 KO cells. Type I IFN response is the first barrier blocking virus invasion in animal cells; increased type I IFN response often indicates the enhanced resistance to viral infection in hosts. qPCR analysis showed the increased IFNβ and IL1β mRNA level in KO cells infected with PRV, indicating that the KO cells harbor more robust type I IFN response than NT cell upon PRV infection (Figure 2G,H). To validate our observations in HDAC6 KO cells, we used the overexpression strategy to see if there were opposite IFN responses to PRV infection. By transfection of PK15 cells with HDAC6-expressing or blank vector (Figure 2I,J) and PRV infection at moi = 0.1 for 24 h, we observed decreased IFNβ mRNA level and increased PRV copies and titers in the HDAC6-overexpressing PK15 cells, clearly demonstrating that HDAC6 facilitates PRV infection by inhibiting type I IFN response in PK15 cells (Figure 2K–M). However, we did not observe increased PRV copies in the HDAC6-overexpressing PK15 cells infected with PRV at moi = 1 (Figure 2N). Since viral infection inhibited HDAC6 expression, high titers of virus can be more robust in reducing HDAC6 expression (Figure 1A–D). We guess that viral infection at moi = 1 may be more robust in compromising overexpressed HDAC6 compared to viral infection at moi = 0.1, thus limiting the IFN-regulatory effect of HDAC6 overexpression. We performed qPCR and WB to compare overexpressed HDAC6 levels between cells infected with PRV at moi = 0.1 and 1 and found a more pronounced reduction in HDAC6 expression at both mRNA and protein levels in PK15 cells infected with PRV at moi = 1 compared to that at moi = 0.1 (Figure 2O,P).

### 3.3. HDAC6 Facilitates VSV Infection by Inhibiting Type I IFN Response in PK15 Cells

We next investigated the HDAC6 effect on RNA virus. We used VSV to infect HDAC6 KO and NT PK15 cells and observed significantly reduced VSV infectivity in all KO cells tested, as evidenced by lower viral copies and TCID50 titers and fewer VSV-positive cells in KO cells than in NT cells (Figure 3A–E). Analysis of the expression of factors in the type I IFN response signal revealed significantly increased IFNβ and IL1β mRNA levels, indicating that HDAC6 KO restricted RNA virus infection in the same manner as DNA virus (Figure 3F,G). This was further confirmed by using VSV to infect HDAC6-overexpressing cells, which resulted in increased VSV copies and decreased IFNβ mRNA levels (Figure 3H–J). Although VSV infectivity was significantly increased in HDAC6-overexpressing cells at moi = 0.1, we did not observe increased VSV copies in the HDAC6-overexpressing PK15 cells infected with VSV at moi = 1 (Figure 3K). This result is consistent with the observation of PRV infection in HDAC6-overexpressing PK15 cells (Figure 2N).

### 3.4. HDACi Antagonizes PRV and VSV Infection in PK15 Cells

Since HDAC6 expression level is directly related with the infectivity of both DNA and RNA virus, treatment with small molecular drugs regulating HDAC expression level could be exploited as the simple strategy to control virus infection. Many HDAC inhibitors (HDACi) have been developed for various therapeutic purposes. We first chose Trichostatin A (TSA), a broad-spectrum HDACi, to test its effect on viral infection in PK15 cells. Cell viability assay showed that TSA lower than 1000 nM in concentration showed no significant toxicity to PK15 cells (Figure 4A). We treated PK15 cells with 400 nM TSA for 12 h, and then infected the cells with PRV at moi = 0.1 for 24 h. Analysis of gene expression showed that 400 nM TSA can greatly reduce HDAC6 mRNA levels and PRV infection further reduced HDAC6 levels (Figure 4B). In contrast, the IFNβ and IL1β mRNA levels were greatly increased upon 400 nM TSA treatment (Figure 4C,D), which is in agreement with the HDAC6 KO effect observed above. As expected, treatment with TSA in the range of 400–800 nM significantly inhibited PRV replication, as evidenced by decreased viral copies and titers in TSA-treated PK15 cells infected with PRV at moi = 0.1 and 1 (Figure 4E,F). TSA treatment also exhibited a similar inhibitory effect on VSV infection in PK15 cells. TSA treatment decreased the HDAC6 mRNA level and increased IFNβ and IL1β mRNA levels, limiting VSV infection in PK15 cells (Figure 4B–D,G,H).

Given the non-specific inhibitory effect of TSA on multiple HDACs, we next used the HDAC6 selective inhibitor, tubastatin A, to test its antiviral effect. Tubastatin A showed no cytotoxicity to PK15 cells at doses below 1000 nM (Figure 5A). When PK15 cells were treated with 100 nM tubastatin A for 12 h and then infected with PRV or VSV at moi = 0.1, significantly decreased HDAC6 mRNA and increased IFNβ mRNA were found in tubastatin A-treated cells compared to untreated cells (Figure 5B,C). Tubastatin A at doses of 100 and 200 nM inhibited infection of both PRV (Figure 5D,E) and VSV (Figure 5F,G) at the moi of 0.1 or 1.

To exclude the direct viral killing effect of HDACi, we mixed 400 nM TSA or 100 nM tubastatin A with PRV and infected PK15 for 1 h at the moi of 0.1 to allow viral attachment and entry into cells. After 1 h, cellular viral DNA was determined by qPCR, and no differences in cellular viral DNA were found between the HDACi-mixed virus and virus-only groups (Figure 5H). This result demonstrated that the inhibitory effect of HDACi on the virus occurs at the viral replication stage and not by directly killing the virus particles.

### 3.5. HDAC6 Modulates DNA Damage Response to Regulate Viral Infection

The underlying cause of HDAC6-mediated type I IFN alteration is worthy of further investigation. Many previous reports have highlighted the role of HDAC6 in linking DNA damage response and viral infection [18,19,20,21]. We used WB to detect γ-H2AX (phosphorylated H2AX at Ser 139), an early and sensitive marker monitoring DNA damage [22]. In non-targeting PK15 cells, PRV and VSV infection increased γ-H2AX levels to 1.6 and 2.4 times, respectively, suggesting that viral infection induces more DNA damage, in agreement with the previous publications [21,23]. When cells were depleted of HDAC6, γ-H2AX levels were reduced by approximately 56% and 75% for PRV and VSV infection, respectively. When HDAC6 was overexpressed, γ-H2AX levels were increased by approximately 16% and 24% for PRV and VSV infection, respectively. Meanwhile, we observed the opposite change in the level of acetyl-H2AX. PRV and VSV infection in HDAC6-KO cells increased acetyl-H2AX by approximately 43% and 13%, respectively, whereas PRV and VSV infection in HDAC6-overexpressing cells decreased acetyl-H2AX by approximately 33% and 22%, respectively, compared to infection in NT cells. The results reflect the deacetylating effect of HDAC6 on H2AX (Figure 6A). The balance between phosphorylation and acetylation modifications of H2AX suggests that HDAC6 deacetylates H2AX to promote its phosphorylation.

To further investigate the true status of DNA damage, we carried out a TUNEL assay to visualize the DNA break sites in the cells [24]. The results showed that viral infection greatly enhanced TUNEL signal shown as bright spots in the nuclei region, compared with mock infection. HDAC6 KO and overexpression increased and decreased the signal of TUNEL-positive spots, respectively, indicating HDAC6 is a DNA damage inhibitor (Figure 6B). The opposite response in γ-H2AX expression and DNA damage status suggests that HDAC6-mediated γ-H2AX levels may not be a true indicator of the DNA damage response. Rather, the changes in γ-H2AX levels were the result of HDAC6-mediated deacetylation. The HDAC6-mediated DNA damage response was also determined by quantifying the abasic (AP) sites, a common form of DNA damage in the genome. The results showed that HDAC6 KO increased the number of AP sites, whereas HDAC6 overexpression decreased the number of AP sites, consistent with the TUNEL results (Figure 6C). Finally, we used endonuclease-mediated genome cleavage to model DNA damage status in cells. The telomere-specific binding protein TRF1 fused to FokI nuclease (TRF1-FokI) can induce DNA double-strand breaks in telomeric chromatin. This method can increase DNA damage and avoid the side effects of chemotherapy and radiation [25]. Compared with the untreated and blank vector-transfected PK15 cells, TRF1-FokI transfection increased IFNβ expression, clearly demonstrating the DNA damage-induced type I IFN production (Figure 6D).

These data indicate that HDAC6 negatively regulates the type I IFN response to promote both DNA and RNA viral infection in pig cells. During infection, viral DNA or RNA can be recognized by host pattern recognition receptors (PRRs) to activate type I IFN-induced cellular antiviral responses. We speculated that the production of host type I IFN was not a single pathway originated by viral nucleic acid recognition, but rather by multiple cellular responses to viral infection. The DNA damage induced by viral infection also produced type I IFN secretion to exert antiviral effect. As the DNA damage inhibitor, HDAC6 reduced the viral infection-mediated DNA damage response, thereby reducing type I IFN secretion to facilitate viral infection (Figure 7).

## 4. Discussion

The present work investigates the role of HDAC6 in viral infection in pig cells. Increased HDAC6 level promotes PRV and VSV infection and HDAC6 ablation limits PRV and VSV infection in PK15 cells. Mechanistic work showed that HDAC6 attenuates the DNA damage response to reduce host type I IFN secretion, thereby promoting viral infection. On the contrary, HDAC6 KO inhibits viral infection by promoting DNA damage-mediated type I IFN secretion. This work demonstrates that HDAC6 could be a potential gene target for modulating viral infection in pigs.

Our work found that both PRV and VSV infection increased DNA damage response. It is reported that viral infection induces host oxidative stress to induce host DNA damages [26,27,28]. TUNEL assay of PRV-infected PK15 cells also showed significant DNA break signals enriched around the CPE area in our results, demonstrating that PRV infection exacerbates DNA damage in the host genome. In addition, some viruses can hijack DNA damage and repair machineries to support viral replication [18,20,21]. γ-H2AX is the common marker of DNA damage [22], but this situation may not always be applicable when cells are affected by HDAC6, because the deacetylase feature of HDAC6 impacts the post translational modification (PTM) status of H2AX. Previous works have demonstrated the balance between phosphorylation and acetylation of H2AX, showing that γ-H2AX focus formation is finely tuned by H2AX acetylation [29,30]. On this basis, HDAC6-mediated alterations of H2AX acetylation influence γ-H2AX levels as well. In HDAC6-deficient cells, increased H2AX acetylation reduced its phosphorylation level and vice versa in HDAC6-overexpressing cells. The WB results validated the balance of the two PTMs of H2AX.γ-H2AX is not truly associated with DNA damage status when HDAC6 level was regulated. As such, we tried to determine the true DNA damage status affected by HDAC6. Multiple lines of evidence confirmed the negative relationship between HDAC6 and DNA damage levels through direct visualization and quantification of DNA damage sites in the genome. HDAC6 overexpression is beneficial for genome DNA integrity in PK15 cells, whereas HDAC6 ablation/suppression induces increased DNA damage sites. The inhibitory effect of HDAC6 on DNA damage response has been reported previously [6,31,32]. Of note, several HDACi or HDAC6i are currently in clinical trials for cancer treatment, clearly indicating HDAC inhibition to induce DNA damage as an efficient genotoxic treatment for cancer therapy [33,34,35,36]. Likewise, HDAC6i or HDAC6 KO, as DNA damage inducers, strengthen the type I IFN response, ultimately limiting viral infection. Taken together, we elucidate a signal pathway by which HDAC6 affects both DNA and RNA virus infection by regulating DNA damage-mediated type I IFN production in pig cells.

Although our work has shown the positive regulation of HDAC6 expression level on viral infectivity, this finding may not be consistent for viruses other than PRV and VSV, as some viruses are type I IFN insensitive or have an inhibitory effect on type I IFN response during infection. It is also noteworthy that other publications have reported the opposite effect to ours on the effect of HDAC6 on the virus. For example, HDAC6 has been reported to inhibit human immunodeficiency virus, Sendai virus, and IAV in different experimental settings, including using HDAC6-overexpressing and KO mouse models [14,37,38,39]. In addition, the changes in HDAC6 expression upon viral infection have been reported to be different. We showed decreased HDAC6 expression upon both PRV and VSV infection in PK15 cells, but a previous publication reported increased HDAC6 expression in several cell lines [21]. More research is needed to understand the inconsistent effect or response of HDAC6. The discrepancy suggests the complicated role of HDAC6 in different cell or animal backgrounds and pathogenic conditions through different mechanisms.

## Figures and Tables

**Figure 1 viruses-17-00090-f001:**
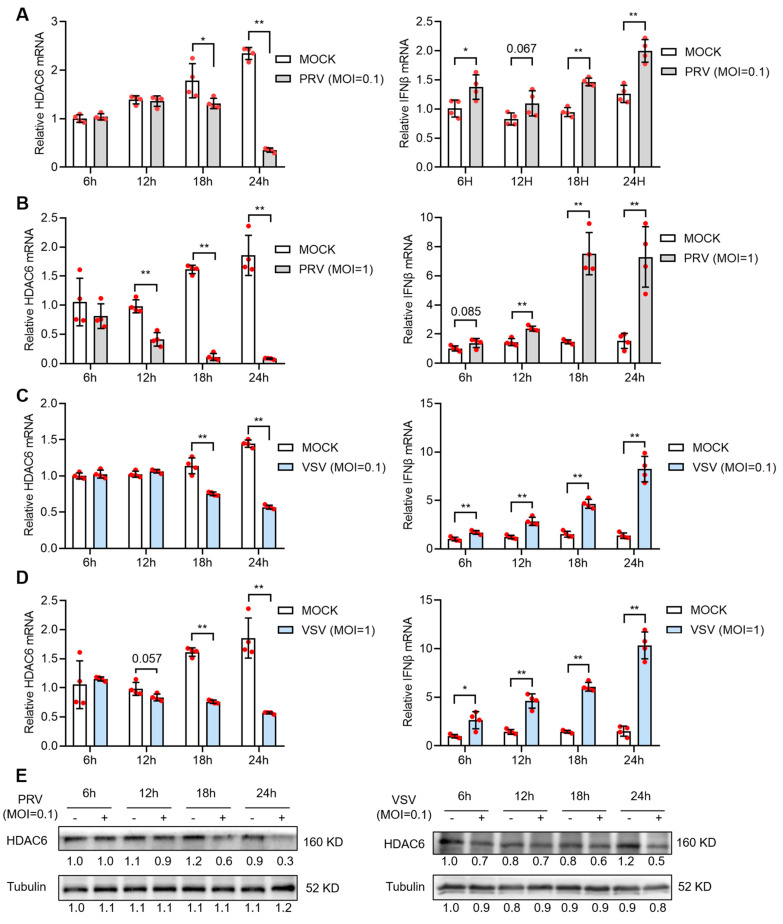
PRV and VSV infection reduces HDAC6 expression in PK15 cells. PK15 cells were infected with PRV (**A**,**B**) or VSV (**C**,**D**) at moi = 0.1 (**A**,**C**) or moi = 1 (**B**,**D**) for 1 h and then refreshed with DMEM for culture. The mRNA expression of HDAC6 and IFNβ was quantified from infected PK15 cells harvested at the indicated time points after infection. Relative expression is normalized to that of mock-infected cells at 6 h. Error bars indicate SD (n = 4). Differences between mock and viral-infection groups were statistically analyzed. Note that (**B**,**D**) share the same mock infection group because they were tested in the same experiment. HDAC6 protein levels at different time points in PK15 infected with PRV or VSV at moi = 0.1 were determined by western blot (**E**). Bands were quantified using ImageJ and normalized to the non-infected group at 6 h. * *p* < 0.05, ** *p* < 0.01, and the insignificant *p*-values close to 0.05 were also marked (two-tailed unpaired Student’s *t*-test for (**A**–**D**)).

**Figure 2 viruses-17-00090-f002:**
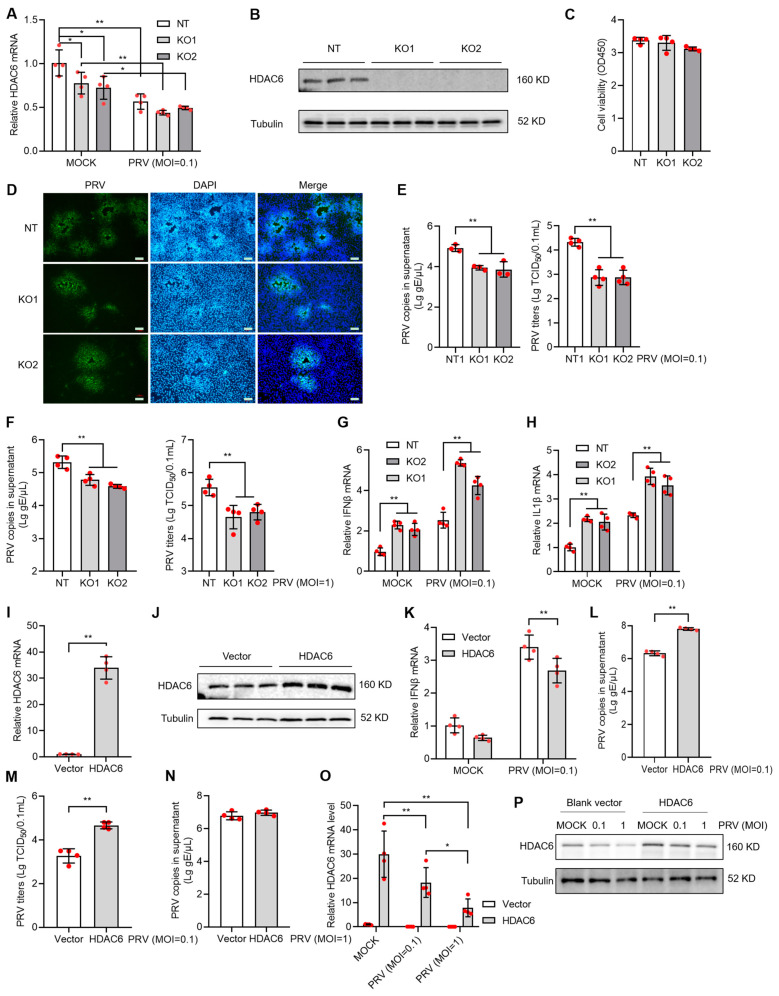
HDAC6 positively regulates PRV infection in PK15 cells. HDAC6 KO cells were screened by CRISPR/Cas9 strategy and the selected single cell colonies were identified by qPCR and WB analyzing HDAC6 mRNA (**A**) and protein (**B**) levels, respectively. Cell viability of KO cells was determined by the CCK8 assay after seeding for 24 h at the same density as NT cells (**C**). The KO and NT control cells were infected with PRV at moi = 0.1. After 24 h, PRV infectivity was assessed by IF assay to label PRV antigen (**D**) in the infected cells. Scale bars = 100 μm. Viral DNA copies and TCID50 titers were determined in the culture supernatant between KO and NT cells infected with PRV at moi = 0.1 (**E**) and 1 (**F**). Mock and PRV-infected NT and KO cells were harvested for total RNA extraction and subjected to qPCR to quantify the mRNA expression levels of IFNβ (**G**) and IL1β (**H**). PK15 cells were transfected with HDAC6-expressing plasmid and blank vector as control. The overexpressed HDAC6 mRNA and protein levels were determined by qPCR (**I**) and western blot (**J**) in the transfecting cells (**J**). After infection with PRV at moi = 0.1 for 24 h, IFNβ mRNA expression in the cells (**K**) and PRV DNA copies in the supernatant (**L**) were determined by qPCR. PRV titers in supernatant were determined by TCID50 (**M**). PRV DNA copies in the culture supernatant were determined after infection with PRV at moi = 1 for 24 h (**N**). HDAC6 expression was determined by qPCR (**O**) and WB (**P**) in HDAC6-overexpressing cells with mock and PRV infection at different MOI. Error bars indicate SD. * *p* < 0.05, ** *p* < 0.01 (two-tailed unpaired Student’s *t*-test for (**I**,**L**–**N**); one-way ANOVA with Holm-Sidak’s multiple comparisons test for (**C**,**E**,**F**); and two-way ANOVA with Holm-Sidak’s multiple comparisons test for (**A**,**G**,**H**,**K**,**O**)).

**Figure 3 viruses-17-00090-f003:**
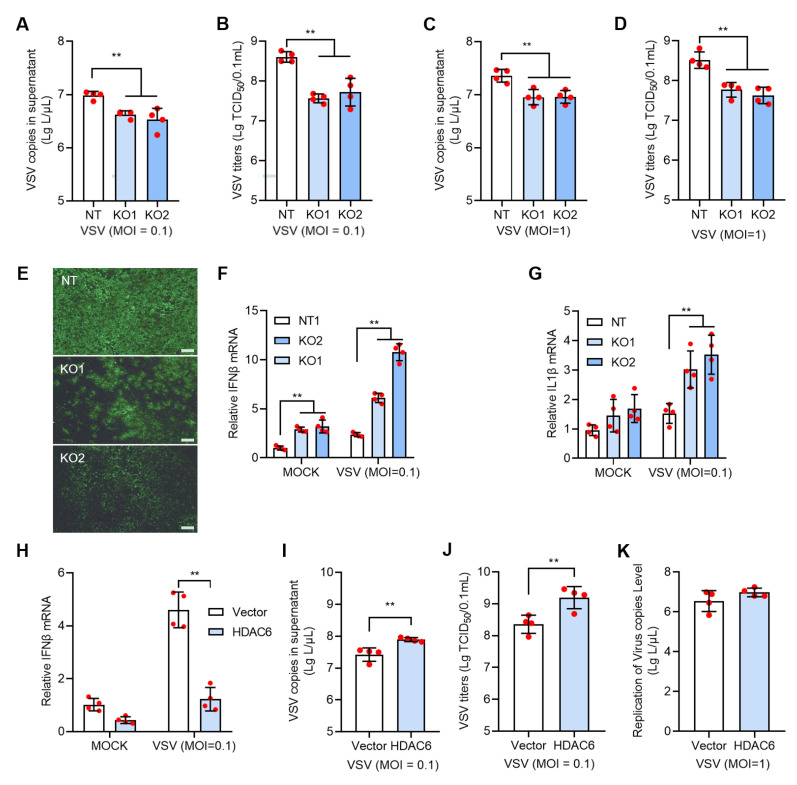
HDAC6 positively regulates VSV infection in PK15 cells. The KO and NT PK15 cells were infected with VSV-EGFP at moi = 0.1 (**A**,**B**) or 1 (**C**,**D**) for 24 h. VSV RNA copies (**A**,**C**) and TCID50 titers (**B**,**D**) were determined in the culture supernatant between KO and NT cells. The representative images of fluorescent cells with VSV-EGFP infection at moi = 0.1 were shown (**E**). Scale bars = 100 μm. The mock and VSV-infected NT and KO cells were harvested for total RNA extraction and subjected to qPCR to quantify the mRNA expression levels of IFNβ (**F**) and IL1β (**G**). The PK15 cells were transfected with HDAC6-expressing plasmid with blank vector as control and infected with VSV at moi = 0.1 for 24 h to analyze the IFNβ expression (**H**) and VSV growth level (**I**,**J**). VSV RNA copies in the culture supernatant were determined after infection at moi = 1 for 24 h (**K**). Error bars indicate SD. ** *p* < 0.01 (two-tailed unpaired Student’s *t*-test for (**I**–**K**); one-way ANOVA with Holm-Sidak’s multiple comparisons test for (**A**–**D**); two-way ANOVA with Holm-Sidak’s multiple comparisons test for (**F**–**H**)).

**Figure 4 viruses-17-00090-f004:**
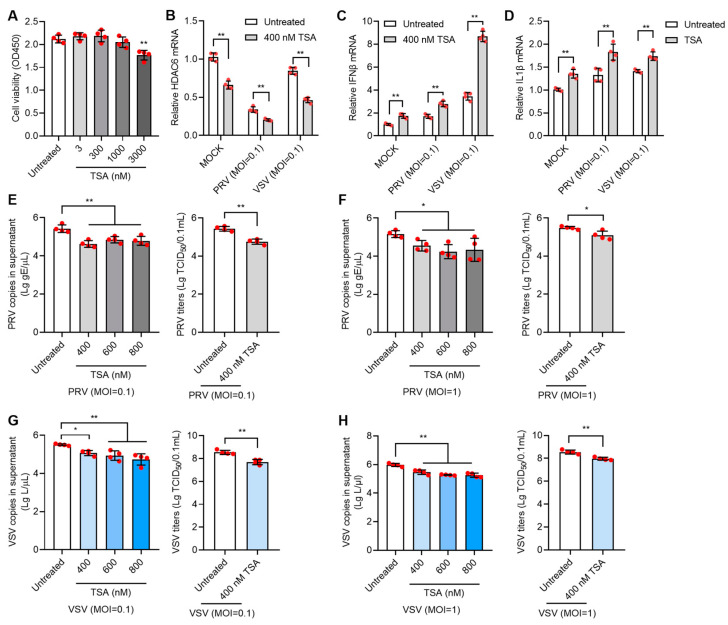
TSA inhibits PRV and VSV infection in PK15 cells. Cell viability assay of PK15 cells treated with 3–3000 nM TSA for 12 h (**A**). PK15 cells were treated with 400 nM TSA for 12 h and then infected with PRV or VSV at moi = 0.1 for 24 h. The infected cells were harvested for total RNA extraction to determine the mRNA expression levels of HDAC6, IFNβ, and IL1β (**B**–**D**). PK15 cells were treated with 400–800 nM TSA for 12 h and then infected with PRV at moi = 0.1 or 1 for 24 h. PRV copies and titers were determined in the culture supernatant (**E**,**F**). PK15 cells were treated with 400–800 nM TSA for 12 h and then infected with VSV at moi = 0.1 or 1 for 24 h. VSV infectivity in TSA-treated PK15 cells was determined by quantifying viral RNA copies and titers in the supernatant (**G**,**H**). Error bars indicate SD. * *p* < 0.05, ** *p* < 0.01 (two-tailed unpaired Student’s *t*-test for (**E**–**H**) (right panels); one-way ANOVA with Holm-Sidak’s multiple comparisons test for (**E**–**H**) (left panels) and (**A**); and two-way ANOVA with Holm-Sidak’s multiple comparisons test for (**B**–**D**)).

**Figure 5 viruses-17-00090-f005:**
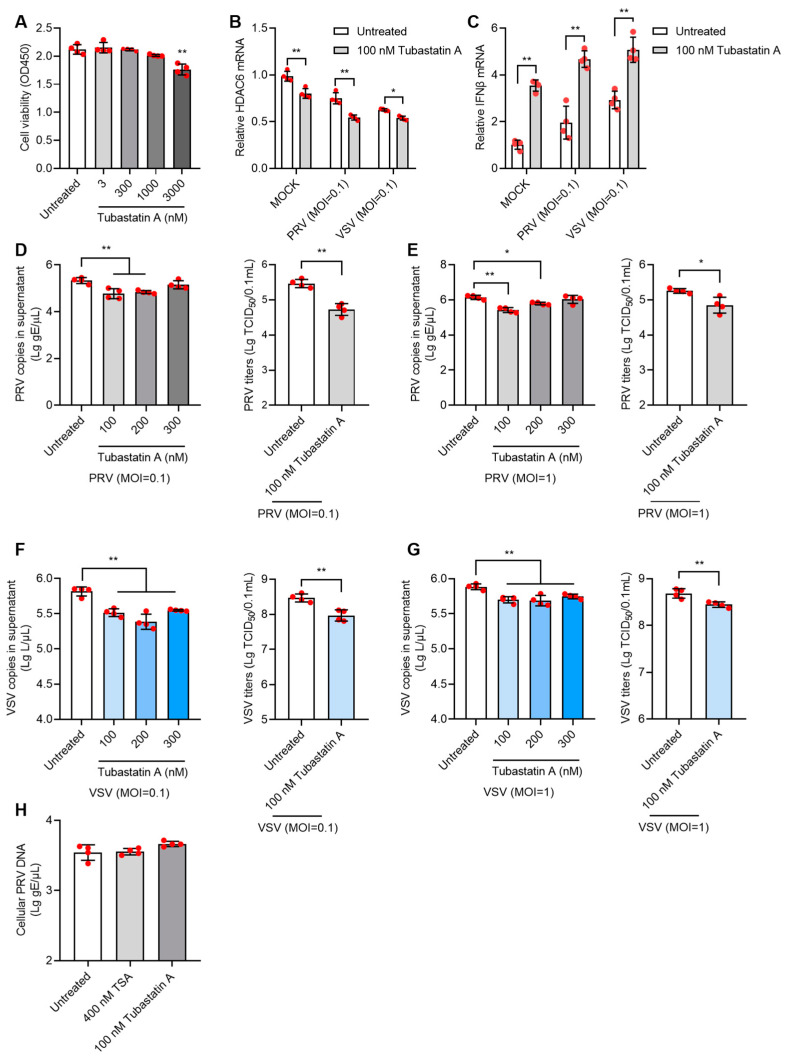
Tubastatin A inhibits PRV and VSV infection in PK15 cells. Cell viability assay of PK15 cells treated with 3–3000 nM tubastatin A for 12 h (**A**). The mRNA levels of HDAC6 and IFNβ in 100 nM tubastatin A-treated PK15 cells infected with PRV or VSV at moi = 0.1 were determined by qPCR (**B**,**C**). PK15 cells were treated with 100–300 nM tubastatin A for 12 h and then infected with PRV at moi = 0.1 or moi =1 for 24 h. PRV genomic copies and viral titers in the culture supernatant were determined (**D**,**E**). PK15 cells were treated with 100–300 nM tubastatin A for 12 h and then infected with VSV at moi = 0.1 or moi =1 for 24 h. VSV genomic copies and viral titers in the culture supernatant were determined (**F**,**G**). To test the direct killing effect of HDACi on virus, PRV was mixed with 400 nM TSA or 100 nM tubastatin A to infect PK15 for 1 h at moi = 0.1. The cellular viral copies which represent virus attachment and penetration level were determined by qPCR (**H**). Error bars indicate SD. * *p* < 0.05, ** *p* < 0.01 (two-tailed unpaired Student’s *t*-test for (**D**–**G**) (right panels); one-way ANOVA with Holm-Sidak’s multiple comparisons test for (**D**–**G**) (left panels), (**A**,**H**); and two-way ANOVA with Holm-Sidak’s multiple comparisons test for (**B**,**C**)).

**Figure 6 viruses-17-00090-f006:**
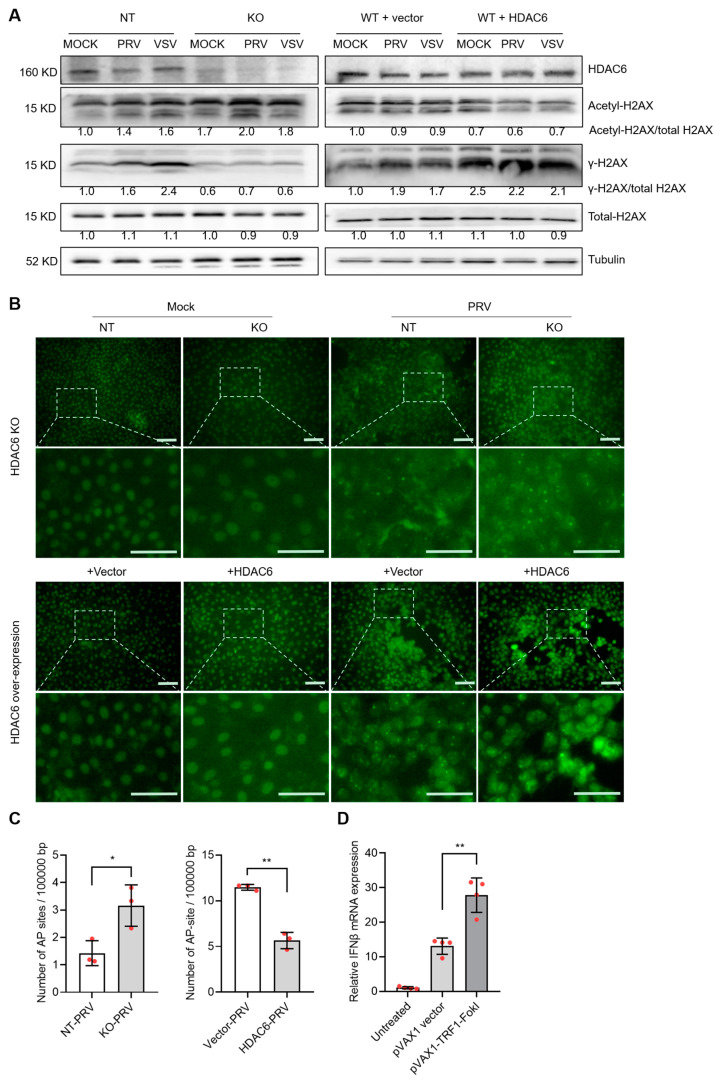
HDAC6 attenuates DNA damage to promote type I IFN production. (**A**) Immunoblot analysis of H2AX acetylation and phosphorylation levels affected by HDAC6 deficiency and overexpression in PK15 cells infected with PRV and VSV at moi = 0.1. The band density of phosphor-, acetyl-, and total-H2AX were quantified and compared. KO and NT represent HDAC6-KO and non-targeting control PK15 cells. WT+vector and WT+HDAC6 represent wild-type PK15 cells transfected with blank vector and HDAC6-overexpressing vector. (**B**) TUNEL assay detecting the DNA break sites in the genomic DNA of HDAC6-KO and overexpressing PK15 cells with mock and PRV infection (moi = 0.1) for 24 h. The insets are amplified to show the green fluorescent labelled dots which represent DNA breaks in the nuclei. Scale bars = 50 μm. (**C**) Quantification of AP sites in the genome of HDAC6-KO and overexpressing PK15 cells infected with PRV at moi = 0.1 for 24 h. (**D**) DNA damage model generated by TFR1-FokI cleavage in the genome of PK15 cells showing DNA damage-induced IFNβ levels. Error bars indicate SDs (n = 3 for (**C**) and 4 for (**D**)). * *p* < 0.05, ** *p* < 0.01 (two-tailed unpaired Student’s *t*-test).

**Figure 7 viruses-17-00090-f007:**
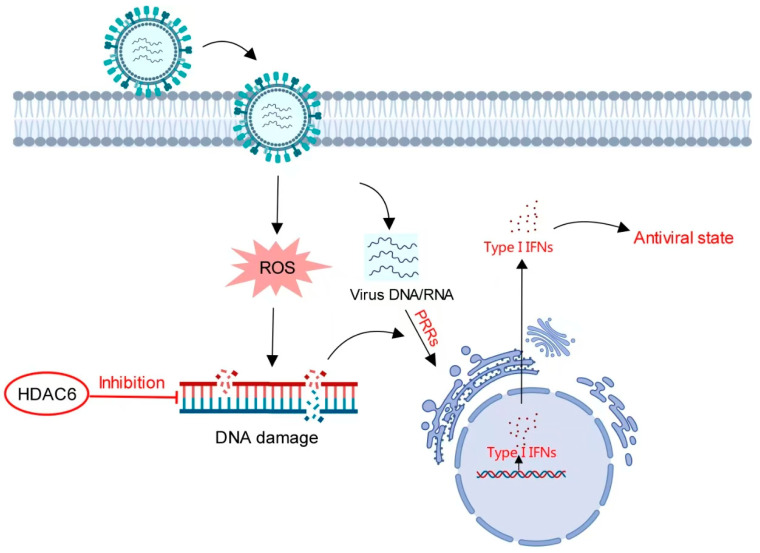
Model for HDAC6-mediated regulation of type I IFN. Viral infection induces increased DNA damage to stimulate type I IFN production in a pathway distinct from the IFN signaling pathway mediated by viral nucleic acid-sensing PRRs. HDAC6 inhibits DNA damage. In the presence of HDAC6, attenuated DNA damage reduces type I IFN production. In the absence of HDAC6, accumulating DNA damage induces more type I IFN production to create an antiviral state in host cells.

## Data Availability

The raw data supporting the conclusions of this article will be made available by the authors on request.

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
