# Peer review of "HDAC6 Facilitates PRV and VSV Infection by Inhibiting Type I Interferon Production"

_viruses, 2025, doi:10.3390/v17010090_

Round 1

Reviewer 1 Report

Comments and Suggestions for Authors

Zheng et al. report that HDAC6 promotes replication of PRV and VSV by inhibiting type I interferon production. 

The data are of interest and mostly of good quality. Not all data are convincing though and I do have some substantial concerns.

Comments:

1. One major concern that I have is some apparent contradictory results between this study and another recent study (PMID: 36951571). In the latter study, it was shown that PRV infection leads to increased HDAC6 protein levels and that HDAC6 increases the DNA damage response (DDR). However, in the current study, the authors show the opposite: PRV infection leads to decreased HDAC6 protein levels and HDAC6 suppresses the DDR. Both studies used PK15 cells and similar infectious doses, making this apparent contradiction even more striking. The authors should explain or at least discuss these apparent contradictory results. Although the author cite the other paper (ref nr 18), they cite it in a very general way (line 397) without mentioning or discussing the apparent contradictory results.  

2. The interaction between HDAC6 and the DDR seems to be complex. The paper creates the perception that HDAC6 in general suppresses the DDR, but there appear to be several papers indicating that HDAC6 activates the DDR, not only the paper mentioned in comment 1 (PMID: 36951571) but also other papers (e.g. PMID: 34584069, 39567688). The authors should nuance their statements regarding HDAC6 and the DDR and indicate that HDAC6 appears to be an important regulator of the DDR but that this may result in either suppression or activation of the DDR, probably depending on the circumstances.

3.  Along the same lines, comment 2 makes the suggestion of the authors that the creation of HDAC6 KO pigs may represent an interesting road towards virus-resistant transgenic pigs less feasible and likely, also since HDAC6 not only increases virus replication but for some viruses has been reported to suppress replication, as also acknowledged by the authors (lines 48-49), so I suggest that the authors soften their suggestions regarding HDAC6 KO pigs or HDAC6-targeting drugs (e.g. lines 20-21, 50-53). They do mention this in the discussion (lines 420-429), but I think it would be good to nuance the statements in the abstract and introduction.

4. Figure 1E: Western blot HDAC6 reduction in the course of PRV infection is not convincing. Mock also shows a reduced HDAC6 signal over time that is comparable to that observed for PRV. Either provide more convincing data or adapt conclusions.

The reduced HDAC6 mRNA levels over time during virus infection are more convincing. For PRV, this is not really surprising seen the broad suppression of host gene expression during PRV infection (PMID: 35730976), the authors should acknowledge/cite this. 

5. Fig 1F & 1G: Authors indicate that UV-inactivated PRV or VSV also triggers reduced HDAC6 mRNA levels. Although this would suggest that virus entry may lead to this reduction in mRNA levels, this does not correspond with the HDAC6 mRNA reduction observed in the time course assays with infectious virus. Using a MOI 1 (so almost all cells are infected), authors observe HDAC6 mRNA reduction only at relatively late time points: later than 6hpi for PRV (Fig 1B) and later than 12hpi for VSV (Fig 1D), which is long after virus entry. So these data are apparently contradictory. Are the authors sure that the UV treatment inactivated the viruses? In addition, the reduction in HDAC6 mRNA using UV-inactivated PRV or VSV is very weak (certainly for PRV compared to infectious virus)(Fig 1F & 1G). I suggest that the authors either remove the UV-treated virus data or substantially nuance their conclusions as I don't think their data suggest that the HDAC6 mRNA reduction occurs (partly) during virus entry.

6. The authors generated HDAC6 knockout (KO) PK15 cells. However, Fig 2B shows that these KO cell lines still express detectable HDAC6 mRNA. How do the authors explain this? Did the crisps/cas9 approach result in the generation of truncated HDAC6 mRNA that is still picked up by their RT-qPCR? Did the authors sequence the KO cell lines to ensure that they are really KO in both alleles?

7. Lines 195-196 and Fig 2D: I do not agree that the HDAC6 KO cells show reduced PRV plaque sizes. Plaques look very similar/identical in size. 

8. Fig 2N (PRV) and Fig 3K (VSV): why is there no difference in viral genome copy numbers in vector-control-transfected versus HDAC6-transfected cells when infecting PRV or VSV at MOI 1 (while there is an effect at MOI 0.1: Fig 2L and Fig 3I)? Is this because the effect is very minor and only leads to detectable differences upon multiple replication rounds (at MOI 0.1) but is not observed upon a single round of infection (MOI 1)?

The authors 'guess' (Fig 247-249) that this may be due because at MOI 1 the viruses may be more robust in compromising (over expressed) HDAC6. This seems easy to test, and would avoid guessing: did the authors check HDAC6 levels in HDAC6-overexpressing cells infected with PRV or VSV at MOI 1 versus MOI 0.1? This easy assay would allow to confirm or deny their hypothesis. 

9. Lines 258-259: HDAC inhibitor TSA led to reduced HDAC6 mRNA levels. Why is that? Since TSA is a broad HDAC inhibitor, I understand that it inhibits HDAC6 activity but why does it have an effect on HDAC6 mRNA levels? 

10. Figure 3F and Figure 3G mock conditions are the exact same conditions as the mock conditions in Figure 2G and 2H. However, in the latter mock conditions (2G and 2H), there are no obvious or statistically significant differences between WT or KO1 or KO2 cells, whereas in the former (3F and 3G) there are significant differences between WT/KO1 and KO2. Can the authors explain these apparent contradictory results?

11. Lines 312-313 and Fig 6A: PRV and VSV trigger increased gamma-H2AX. At least for PRV, this has been reported before and should be acknowledged, e.g. PMID 34613805, 36951571.

12. I do not agree with the conclusion/statement that HDAC6 attenuates the DDR (line 307 and throughout the manuscript). The data (Figure 6A) show that in the absence of HDAC6 (KO), there is reduced H2AX phosphorylation (a hallmark of DDR), which indicates that HDAC6 promotes the DDR rather than attenuating it. Although I agree that the (opposite) results of acetylated HDAC6 should also be taken into account, I think the authors cannot conclude that HDAC6 suppresses the DDR - they can conclude that HDAC6 modulates the DDR.

Comments on the Quality of English Language

Use of the English language is certainly not bad, but could be improved to remove inaccuracies and increase clarity. I suggest that the authors let a native English speaker carefully edit the manuscript.  

Author Response

We have substantially revised the manuscript according to the reviewers' comments, and the major revisions have been highlighted in colour in the revised manuscript. The point-by-point response to the reviewers' comments has been uploaded as an attachment below.

Reviewer 2 Report

Comments and Suggestions for Authors

Introduction:

1. The authors only introduced the background of HDAC6 but no PRV or VSV and why they wanted to target these two viruses.

2. I wish to see more introduction about HDAC6 in the regulation of gene expressions as the viral/host expressional changes may lead to cellular survival or death.

3. They should mention the full names of PRV and VSV whenever they mention them for the first time in the article. 

Results:

1. If the cytotoxicity of Tubastatin A is below 1000 nM and the authors did not see a significant difference when using 300 nM, why did they not choose higher concentrations?

2. 5D and 5E, it is hard to explain why Tubastatin A could not reduce the viral RNA when using 300 nM, which was a relatively higher concentration than 100 and 200 nM.

Overall

I wonder the reudction of VSV and PRV roduction:

1. The authors only introduced the background of HDAC6 but no PRV or VSV and why they wanted to target these two viruses.

2. I wish to see more introduction about HDAC6 in the regulation of gene expressions as the viral/host expressional changes may lead to cellular survival or death.

Results:

1. If the cytotoxicity of Tubastatin A is below 1000 nM and the authors did not see a significant difference when using 300 nM, why did they not choose higher concentrations?

2. 5D and 5E, it is hard to explain why Tubastatin A could not reduce the viral RNA when using 300 nM, which was a relatively higher concentration than 100 and 200 nM.

Overall

I wonder if the reductions of VSV and PRV were due to the slower cell proliferation when cells were treated with HDAC inhibitors. After all, the viral inhibition of TSA or Tubastatin A was very limited and not as significant as we expected. Lower cell proliferation may lead to slower viral replication as well. Also, PRV is a DNA virus that may require HDAC when interacting with histones in the nucleus, but VSV, an RNA virus, doesn't enter the nucleus. Could it be beneficial for PRV to stay in the nucleus when adding those inhibitors? 

Round 2

Reviewer 1 Report

Comments and Suggestions for Authors

The authors addressed my concerns.